# Associations between vision impairment and driving and the effectiveness of vision-related interventions: protocol for a systematic review and meta-analysis

Helen Nguyen ![ORCID],[1] Gian Luca Di Tanna,[2] Kristy Coxon ![ORCID],[3] Julie Brown,[2] Kerrie Ren,[1] Jacqueline Ramke ![ORCID],[4,5] Matthew J Burton,[4,6] Iris Gordon,[4] Justine H Zhang ![ORCID],[4,7] João M Furtado,[8] Shaffi Mdala,[9] Gatera Fiston Kitema,[10] Lisa Keay[1,2]

► Prepublication history an additional material for this paper is available online. To view these files, please visit the journal online (http://dx.doi.org/10.1136/bmjopen-2020-040881).

For numbered affiliations see end of article.

**Correspondence to**
Professor Lisa Keay;
l.keay@unsw.edu.au

## ABSTRACT

**Introduction** Driving is one of the main modes of transport with safe driving requiring a combination of visual, cognitive and physical skills. With population ageing, the number of people living with vision impairment is set to increase in the decades ahead. Vision impairment may negatively impact an individual's ability to safely drive. The association between vision impairment and motor vehicle crash involvement or driving participation has yet to be systematically investigated. Further, the evidence for the effectiveness of vision-related interventions aimed at decreasing crashes and driving errors has not been synthesised.

**Methods and analysis** A search will be conducted for relevant studies on Medline (Ovid), EMBASE and Global Health from their inception to March 2020 without date or geographical restrictions. Two investigators will independently screen abstracts and full texts using Covidence software with conflicts resolved by a third investigator. Data extraction will be conducted on all included studies, and their quality assessed to determine the risk of bias using the Joanna Briggs Institute Critical Appraisal Tools. Outcome measures include crash risk, driving cessation and surrogate measures of driving safety (eg, driving errors and performance). The results of this review will be reported using the Preferred Reporting Items for Systematic Review and Meta-Analysis guideline. Meta-analysis will be undertaken for outcomes with sufficient data and reported following the Meta-analyses of Observational Studies in Epidemiology guideline. Where statistical pooling is not feasible or appropriate, narrative summaries will be presented following the Synthesis Without Meta-analysis in systematic reviews guideline.

**Ethics and dissemination** This review will only report on published data thus no ethics approval is required. Results will be included in the *Lancet Global Health* Commission on Global Eye Health, published in a peer-reviewed journal and presented at relevant conferences.

**PROSPERO registration number** CRD42020172153.

### Strengths and limitations of this study

- ► Results from this systematic review will present up-to-date evidence for the influence of vision impairment on road traffic injuries and the effectiveness of vision-related interventions.
- ► As there are no geographic restrictions in the criteria for included studies, this review will capture a large portion of English-language publications in this research area with findings applicable to a global context.
- ► This review will not restrict the age of the target population allowing evidence on the impact of vision impairment and driving to be documented for all age groups.
- ► This review only looks at published studies in English, so research from non-English speaking countries will be missed, which could introduce bias.
- ► Another potential limitation is that interventions and outcome measures may be highly heterogeneous, which will affect the conclusions drawn from the results and prevent meta-analyses to be conducted for select outcomes.

## INTRODUCTION

According to the WHO,[1] approximately 1.35 million people die each year from road traffic injuries (RTIs), making it the eighth leading cause of death globally. Low-income and middle-income countries (LMICs) have lower rates of vehicle ownership compared with high-income countries (HIC), but over 90% of RTI fatalities occur in LMICs with the highest death rates in Africa. Individuals from lower socioeconomic backgrounds living in HICs are also more likely to be involved in a road crash resulting in injuries. RTIs make up a major proportion of a country's economic and social burden,[2–4] and account for almost

30% of global injury-related disability.[2] In the face of increasing motorisation, achieving absolute reductions in RTIs is a challenge, especially for vulnerable road users such as pedestrians and users of powered two and three wheeler vehicles. This challenge has a direct impact on the UN's Sustainable Development Goals, in particularly Target 3.6 which called for a halving of global road deaths by 2020, and Target 11.2 which called for safe and sustainable transport systems, especially for vulnerable road users.[5]

Motor vehicle crashes (MVCs) and, by extension, RTIs, however, are preventable. Using the Haddon Matrix, an early theory describing the multifactorial nature of RTIs, MVCs are understood to involve host (human), agent (vehicles and equipment) and environmental (physical and socioeconomic) factors.[6] This theory has since been used to build the Safe System Approach endorsed in the United Nations Road Safety Collaboration's Decade of Action for Road Safety (2011–2020).[7] In brief, the Safe System Approach aims to prevent MVCs which result in serious injuries or death by addressing four main pillars of focus: (1) safe roads, (2) safe speeds, (3) safe people and (4) safe vehicles.[8] Road safety programmes, such as the Bloomberg Initiative for Global Road Safety (2015–2019), focus on improving road safety through legislation in LMICs,[9] thus addressing the environmental and agent risks of RTIs. Beyond road infrastructure and vehicle quality, human driving behaviours or 'human factors' also contribute to RTI rates and are an intrinsic part of the Safe System Approach. Safe driving requires individuals to have a range of physical, visual and cognitive skills. In addition to specific eye diseases, age-related functional declines across a range of domains, including vision, can reduce confidence in driving ability.[10] Poor visual acuity and contrast sensitivity, visual field loss and glare sensitivity have all been identified as potential factors contributing to poor driving performance and increased MVCs.[11]

Due to the high visual demands needed to drive safely, many countries have set federal-specific and/or state-specific standards for vision, mostly for visual acuity. Most countries accept that a visual acuity of at least 6/12 (0.50, 20/40) in the better eye is considered as the requirement for driving. This threshold dictates jurisdictional control used to identify individuals with vision impairment and restrict their access to driving privileges. However, a systematic review by Dobbs suggested that licencing policies aimed at identifying at-risk older drivers may not be effective in decreasing crash rates.[12] This may be because policies which govern licensure and vision screening vary significantly between and within countries. Further, evidence on their effectiveness is inconclusive.[13] Conversely, in-person renewal policies, which include vision tests completed at licence renewal centres, have been shown to reduce crash rates in older drivers.[14] An American study analysing data from the National Highway Traffic Safety Administration Fatal Accident Reporting System found drivers aged 70 years and older who underwent visual acuity examinations during their licence renewals had lower fatal crash risks than their non-vision tested peers (risk ratio (RR) 0.93; 95% CI 0.89 to 0.97).[15] However, the literature remains divided in its support for using visual acuity alone as a predictor of MVC involvement and high-risk driving behaviours.[16] A Cochrane review, updated twice, examined the benefits of different vision screening procedures, such as visual acuity, visual field (central or peripheral), contrast sensitivity and useful field of view tests, in randomised controlled trials (RCTs) aimed at preventing RTIs and fatalities in older drivers.[13 17] Unfortunately, no RCTs met the inclusion criteria for the review at the time these reviews were conducted.

There is substantial literature investigating how vision impairment, and other aspects of function, affect road safety. Measures of driving safety have included indirect measures such as performance on driving simulators, on-road driving assessments, naturalistic driving or in-vehicle monitoring as well as direct measures of RTI and MVC rates from self-report or administrative datasets.[18 19] However, the evidence for the influence of vision loss on MVCs and the corresponding benefits of interventions to restore vision have not been systematically evaluated. Since older drivers have higher crash involvement[20] and greater prevalence of eye diseases due to natural age-related declines in vision,[21] most research investigate older drivers and their risks of crashes and injuries. However, it is important to document the impact of vision impairment across all age groups. Further, information is also needed about specific eye diseases and types of vision impairment to inform interventions to screen for poor vision in drivers, and interventions to rehabilitate vision, thereby enhancing driver safety and continued ability to drive. This is especially important for older adults who rely on driving to remain independent and connected with their community. The loss of the ability to drive and the eventual retirement from driving has been linked to higher symptoms of depression and poorer health in older adults.[22]

The aim of this systematic review is to: (1) describe the associations between vision impairment and risk of road crash involvement across the lifespan, and (2) evaluate vision-related interventions to reduce crash risk. Secondary outcomes are driving cessation and surrogate measures of crash risk such as on-road driving errors.

## METHODS AND ANALYSIS

This systematic review protocol was drafted using the International Prospective Register of Systematic Reviews (PROSPERO) as a guideline and registered in PROSPERO (28 April 2020 (https://www.crd.york.ac.uk/prospero/display_record.php?ID=CRD42020172153)). Any changes to the protocol will be updated in PROSPERO. The protocol is prepared in accordance to the Preferred Reporting Items for Systematic review and Meta-Analysis Protocols statement (online supplemental appendix 1).[23]

## Eligibility criteria

This review will include human studies in the English language with full text available. Unlike the two previous Cochrane reviews which only included RCTs, this review will consider both interventional (RCTs and quasi-experimental) and observational (cohort, cross-sectional and case–control) studies. Systematic reviews will be included if meta-analysis was performed. For systematic reviews without meta-analysis, the reference list will be examined for potentially relevant articles, but the systematic review itself will not be included. All literature reviews, commentary articles, dissertations, abstracts, editorials and conference presentations will be excluded.

All studies must report on at least one of the outcome variables, described in the following section, which include MVC involvement and surrogate measures of driving safety such as driving errors and performance scores and driving cessation. Studies investigating either self-regulatory behaviours, such as night driving avoidance and decreasing travel mileage, or self-reported measures of driving safety will be excluded. To obtain data on driving scores and performance, studies using on-road driving tests, which include closed-circuit routes and those combining both closed and real-road driving tracks, and naturalistic driving with in-vehicle monitoring will be included. Even though closed circuits may not reflect true on-road driving conditions, tests for common driving maneuverers such as road signage recognition, hazard recognition and avoidance, reversing and gap perception are able to be recreated on these routes.[24] Driving errors and driving performance scores on the on-road driving tests can come from fitted in-vehicle monitoring technologies or trained observers. To restrict the scope of the study to direct measures of driving, studies which used driving simulators will be excluded as these are laboratory studies with only indirect measures of driving performance. MVC involvement cannot be measured and real-life driving experiences, such as limited exposure to driving at night, in bad weather or during rush hour, may not be reflected in a simulation.[25] Additionally, the validity of driving simulator results are highly dependent on the type of simulation programme used and what kind of driving manoeuvre is being investigated.[26] As this review is interested in the MVC involvement and driving abilities of individuals who drive and their habitual vision, studies which simulate impairments in vision will also be excluded.

The population of focus will be drivers of four-wheeled motorised vehicles such as cars, buses, and trucks. Unlike the two Cochrane reviews mentioned above, which only focused on older drivers, this review will include drivers of all ages. Studies of drivers who have specific medical conditions (eg, dementia, epilepsy, stroke and history of medical events such as syncope), or vision difficulties due to other medical factors (eg, hemianopia caused by brain damage) will not be included. Similarly, articles where vision status is not reported will be excluded.

Exposures in the included studies will encompass all types of vision impairment including visual acuity, contrast sensitivity, visual field loss as well as impairments associated with specific eye diseases including but not limited to glaucoma, cataracts, aged-related macular degeneration, diabetic retinopathy, stereopsis disorders and colour vision deficiencies. Vision impairments can be categorised by the specific eye diseases or by specific measures of vision which can negatively impact normal everyday functioning. Even though it is not necessary for all included studies to report on vision-related interventions, studies which do report on interventions can include procedures such as vision screening, refractive correction, cataract surgery or other measures to restore and improve vision of drivers in order to maintain driving participation, promote safe driving and reduce risk of crash involvement. The exposure comparators of included studies will be drivers who either do not have a vision impairment or have not received a vision-related intervention, within a timeframe chosen by the study in question.

## Outcome measures

The primary outcome measure is MVC involvement including fatal MVC involvement. Data on crash involvement and its severity can either come from self-reported surveys or data linkage with government and/or hospital records. Data from self-reported surveys will ensure that MVCs which were not serious enough to warrant a police or hospital report will be also be included.

Driving cessation and surrogate measures of driving safety will be the secondary outcomes. The surrogate measures of driving safety can include scores of driving performance from on-road driving tests or 'naturalistic' in-vehicle monitoring looking at manoeuvres such as lane keeping, braking and abidance of road signage like traffic lights, stop and give way signs. To account for differences in the criteria used by trained observers to evaluate the driving performance scores on on-road driving tests, a pass/fail threshold for driving performance scores specific to this review will be decided on by all investigators in order to synthesise results.

## Search strategy

Electronic database search will be conducted by the Cochrane Eyes and Vision Information specialist (IG) on Medline (Ovid), EMBASE and Global Health from their inception to March 2020. Online supplemental appendices 2–4 show the search strategies for Medline, EMBASE and Global Health, respectively. Additional potentially relevant studies will be sought by experts in the field by checking the reference lists and citations of included studies, and checking the reference list of narrative systematic reviews identified in the search.

## Data collection and analysis
### Data management and selection

Each title and abstract will be screened by two investigators independently (from HN, KR, JR, JHZ, SM, JF, GFK)

using Covidence systematic review management software (Veritas Health Innovation, Melbourne, Australia; available at https://www.covidence.org/home). Full-text review of potentially relevant articles will then be conducted by two investigators independently. Discrepancies will be discussed and resolved via consultation with a third investigator.

### Data extraction

Data extraction will be completed independently by two investigators (from among the same seven investigators). Data from included studies will be extracted using adaptions of the Joanna Briggs Institute (JBI) template for systematic reviews and observational studies (including cohort, cross-sectional and case–control studies).[27] Adapted Cochrane templates will be used to extract data from randomised controlled trials and quasi-experimental studies.[28]

### Quality assessment

A quality assessment to determine an overall risk of bias will be carried out on all included studies independently by two investigators (from the seven investigators mentioned previously). Conflicts will be resolved by a third investigator. Relevant JBI critical appraisal tools will be used to evaluate randomised controlled trials, quasi-experimental studies, systematic reviews, cohort studies, cross-sectional studies and case–control studies.[29]

### Data synthesis strategy

Measures of association between vision impairment/vision-related interventions and MVC involvement, driving cessation and surrogate measures of driving safety will be summarised according to the outcome measures reported in the primary studies. In particular, appropriate HR, RR and OR for binary data and (standardised) mean differences for continuous data will be statistically pooled. When the same outcome is reported as dichotomous data in some studies and as continuous data in others, these studies will be pooled by expressing the ORs as standardised mean differences and vice versa.[30] P values of the driving outcomes will also be reported where appropriate.

Where it is not possible or suitable to statistically pool the studies, a narrative summary of the findings will be used instead. Narrative summaries will follow the Synthesis Without Meta-analysis reporting guidelines.[31] Heterogeneity across all included studies with sufficient data will be assessed clinically, methodologically and statistically. Clinical heterogeneity will be assessed by comparing the differences between the participant characteristics (eg, age, sex, eye disease, driving mileage, licence status or other available measures of driving exposure), interventions and outcomes measured. The design and quality of included studies will be compared to assess methodological heterogeneity. Statistical heterogeneity across studies will be explored by formal statistical test of heterogeneity, subgroup analyses and, if feasible, by meta-regression.

Inconsistency of the effect sizes across the studies will be assessed by the proportion of variability in the effect sizes of the included studies due to heterogeneity (and not by sampling error) using $I^2$. Estimates will be pooled using random effects models with fixed-effect models, results also reported regardless of the values of $I^2$, and prediction intervals to allow for expected effects of future studies to be extrapolated based on the current evidence.[32]

The following outcomes will be assessed using meta-analysis where feasible according to data availability: crash involvement, driving cessation and surrogate measures of unsafe driving, that is, driving errors and driving performance. Furthermore, meta-analyses for each of the different eye diseases, and studies from LMIC settings will also be performed independently for each outcome of interest if possible. As there is no age restriction on the focus population, results on age will be synthesised by assessing specific subgroup analysis and/or meta-regression, which may partially explain heterogeneity across studies in the pooled effect size. The Meta-analysis of Observational Studies in Epidemiology guidelines will be used to guide reporting.[33] The Grading of Recommendations Assessment, Development and Evaluation approach will be used to assess the quality of evidence in the meta-analyses.[34]

Sensitivity analysis will be performed on low risk of bias studies, while the meta-analysis will include all studies. This will assist with verifying the strength of the study findings and to assess how different methodologies, sample sizes and statistical analyses have affected this study's results. Furthermore, funnel plots will be used to assess publication bias.

Corresponding authors from publications dated 2010 onwards with missing data of potential use will be approached via email, up to a maximum of three attempts, to request further information. Any unobtainable data will be noted alongside all attempts to contact the respective authors. Even though only available data will be used for the meta-analysis, the effects of any missing data will be considered and their effects discussed in the overall final review.

### Patient and public involvement

This review will only be looking at the existing published literature. No patient or public involvement is currently planned for the design and execution of this review, however public participation may be sought for this review's dissemination.

## ETHICS AND DISSEMINATION

As this review will only be focusing on the currently published literature, ethics approval is not required. Results from this systematic review will be published in an open peer-reviewed journal and will form part of the ongoing *Lancet Global Health* Commission on Global Eye Health.[35] Where relevant, it will also be presented at conferences.

## DISCUSSION
### Significance of this review

The findings of this systematic review may influence future road safety and licencing policies on driving for drivers

with vision impairment. By understanding the visual factors contributing to MVCs, vision-related screening tests for licencing may be reconsidered and updated to increase relevance to driving safety. As mentioned previously, most reviews on driving with vision impairment have been limited to older drivers and the effects of different licencing renewal procedures on their ability to drive. Even though older drivers are at higher risk,[21] this review will seek to capture data on driving and vision impairment for all age groups.

The eligibility criteria for included studies for this review will ensure that global data on vision and driving will be captured. Currently, MVC-related societal burdens and injury-related disability burdens in LMICs are poorly understood, which may partially explain why cost-effective interventions in these countries are rarely undertaken.[2] LMICs tend to focus on legislative interventions, followed by education/training workshops, public awareness campaigns, enforcement measures, speed control and infrastructure improvements.[36] Current data on human factors specifically related to vision impairment in LMICs reported in this review may inform future evidence-based policies on licencing and/or screening policies to address these gaps.

Results from this review may also provide additional evidence on the impact of eye disease-specific interventions on quality of life factors, especially those related to driving and the ability to drive. Interventions to improve and optimise vision are needed for drivers, in recognition of the importance of continued safe driving. This greater awareness in turn will also provide evidence for policies around road safety for individuals with vision impairments.

**Author affiliations**
¹School of Optometry and Vision Science, Faculty of Science, University of New South Wales, Sydney, New South Wales, Australia
²The George Institute for Global Health, Faculty of Medicine, The University of New South Wales, Sydney, New South Wales, Australia
³School of Health Sciences, and the Translational Health Research Institute, Western Sydney University, Penrith, New South Wales, Australia
⁴International Centre for Eye Health, London School of Hygiene & Tropical Medicine, London, United Kingdom
⁵School of Optometry and Vision Science, University of Auckland, Auckland, New Zealand
⁶Moorfields Eye Hospital, London, United Kingdom
⁷Manchester Royal Eye Hospital, Manchester, United Kingdom
⁸Division of Ophthalmology, Ribeirão Preto Medical School, University of São Paulo, Ribeirão Preto, São Paulo, Brazil
⁹Opthalmology Department, Queen Elizabeth Central Hospital, Blantyre, Malawi
¹⁰Ophthalmology Department, School of Health Sciences, University of Rwanda, Kigali, Rwanda

**Contributors** HN, LK and JR conceived the idea for the review. HN drafted and revised the protocol with suggestions from LK, JR, JB, KC, GLDT, KR, MJB, JHZ, IG, JF, SM and GFK who reviewed the protocol and provided feedback on the draft. IG constructed the search. The final version of the manuscript was approved by all authors.

**Funding** MJB is supported by the Wellcome Trust (207472/Z/17/Z). JR is a Commonwealth Rutherford Fellow, funded by the UK government through the Commonwealth Scholarship Commission in the UK. The Lancet Global Health Commission on Global Eye Health is supported by The Queen Elizabeth Diamond Jubilee Trust, The Wellcome Trust, Sightsavers, The Fred Hollows Foundation, The SEVA Foundation, The British Council for the Prevention of Blindness and Christian Blind Mission.

**Competing interests** None declared.

**Patient consent for publication** Not required.

**Provenance and peer review** Not commissioned; externally peer reviewed.

**ORCID iDs**
Helen Nguyen http://orcid.org/0000-0002-5285-1507
Kristy Coxon http://orcid.org/0000-0003-4820-0397
Jacqueline Ramke http://orcid.org/0000-0002-5764-1306
Justine H Zhang http://orcid.org/0000-0001-8385-2003

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
