## [Reviewer comments · BMJ Open]

ARTICLE DETAILS

TITLE (PROVISIONAL)	Associations between vision impairment and driving and the effectiveness of vision-related interventions: protocol for a systematic review and meta-analysis
AUTHORS	Nguyen, Helen; Di Tanna, Gian Luca; Coxon, Kristy; Brown, Julie; Ren, Kerrie; Ramke, Jacqueline; Burton, Matthew J; Gordon, Iris; Zhang, Justine; Furtado, João; Mdala, Shaffi; Kitema, Gatera Fiston; Keay, Lisa

VERSION 1 – REVIEW

REVIEWER	Danielle McCartney University of Sydney, Australia
REVIEW RETURNED	01-Jul-2020

GENERAL COMMENTS	This protocol describes the method of a systematic review and meta-analysis that will investigate the relationship between vision impairment and MVC involvement, as well as the effectiveness of certain interventions to reduce the rates of MVC involvement among vision-impaired drivers. The question is interesting and worthy of consideration. In addition, the study has a number of strengths – including its use of an on-road driving test. However, several important points remain to be clarified. Title: • I find the “of associations” part of the title to be unclear. I would suggest rephrasing. Abstract: • “Driving is becoming one of the main modes of transport” – is ‘becoming’ accurate here? Is it not already a very popular mode of transport? ‘However’ is also used incorrectly in this sentence, e.g. “with safe driving requiring a combination of...” Strengths and Limitations Section: • The first dot point is quite wordy. I would suggest refining this.• I would be cautious about considering the lack of a time constraint on your search as a strength. It is helpful in terms of providing a complete picture of the evidence; however, public health interventions and driving conditions have evolved a lot over the years and so findings from early studies might have limited relevance today.• The second dot point is also quite vague. Increased data capture is obviously important – you need to focus on the implications of this (e.g. generalisability of findings). You could also incorporate something about the benefits of including all age brackets here.• Will this review really exclude studies that are not published open access, or has this been described incorrectly? This method would
--

result in a lot of missing data and be a significant limitation of the review. Please clarify.

Introduction:

The introduction is informative and generally well written.

However, I have noted a few minor issues below:

- In the first paragraph, the authors describe the prevalence of RTI fatalities in LMIC. If I understand correctly, this has been done to provide insight into the extent of the issue and highlight the fact that RTI is a global problem. I appreciate the detail; however, it leaves the reader with the expectation that the review is going to focus on LMIC. I would suggest removing the third sentence about 'growing disparities' and perhaps replacing this with some additional detail about the state of the problem in high-income countries (even if it is less severe and improving).
- I would suggest not starting the second sentence with the phrase "it has long been established".
- I am not sure the second sentence of the second paragraph makes sense. How has the knowledge that MCVs are multifactorial evolved to the Safe System Approach? I would suggest rephrasing.
- The first sentence of the third paragraph is missing some detail. The authors state that "standards [mostly for visual acuity] have been set" – can you clarify, in which regions are these enforced? (Or, if they are global, state this?). Also, based on your follow up sentences, I would assume that different regions use different methods to identify vision-impaired drivers. Could you perhaps note this (even just subtly in the way the first or second sentence is structured)? It might help to provide context and also introduce the fact that the effectiveness of the different interventions appears to vary and therefore warrants investigation.
- Could you perhaps give some examples of different vision screening procedures toward the end of the third paragraph?
- I find it a bit unusual that the abstract emphasises the need for effective vision-related interventions due to the aging population, but that the aging population isn't considered in the introduction. Is this worthwhile noting?

Method:

- The authors state that studies using driving simulators will be excluded. I assume similar studies using on-road assessment techniques will also be excluded. Perhaps state this outright.
- Studies will obviously have to have measured the primary outcome variable (MVC involvement) to be included in this review. I wonder whether this is worth noting in the 'Eligibility Criteria' section of your paper (and then referring the reader to the 'Outcome Measures' section of the manuscript for further detail)? I think you could use this information to strengthen your justification for excluding driving simulator studies. Currently, your comments on these studies appear to be tacked on the end of a paragraph about study design. I think it would be more appropriate to indicate that eligible studies had to have measured the primary outcome variable and that those driving simulator and on-road studies were excluded because they are not suitable for measuring MVC involvement, specifically? It would also be a more appropriate place to note that you did not intend to include studies investigating "self-regulatory behaviours" (again, specific outcomes).
- The authors plan to exclude studies of drivers with "specific medical conditions" and give examples of some conditions, most

	of which would significantly impact an individual’s ability to safely operate a motor vehicle. I wonder how they intend to handle studies of drivers with less overtly impairing medical conditions (e.g. diabetes)  • Check grammar on Page 7 Line 33 (abstract or abstracts?). • The authors state that “heterogeneity for all included studies will be assessed clinically, methodologically and statistically”. Can heterogeneity be assessed statistically for all included studies? If I understand correctly, this means that all included studies will need to have reported sufficient data to facilitate the completion of these analyses. If this is the case, it needs to be included as an eligibility criterion. • I see now that the authors plan to meta-analyse several different outcomes. My initial interpretation was that MVC involvement was the only outcome that would undergo meta-analysis, since systematic reviews often draw their secondary outcomes only from studies that report the primary outcome – making them unsuitable for meta-analysis (i.e. since some data is not captured). It would be helpful if the authors could specify exactly what a study needs to have measured in order to be eligible for review. Discussion:  • Please review your grammar in the first sentence of the discussion; I’m not sure how this systematic review will care for people who would like to drive. • Please review the wording of the second sentence of the discussion. I think ‘identifying’ is the wrong term here. Perhaps “gaining a better understanding of visual factors that influence crash involvement” (or similar)? • Again, please review the wording of the last paragraph of the discussion. This doesn’t seem to be particularly well-written/considered.
--	--

REVIEWER	Garrett Swan Schepens Eye Research Institute of Massachusetts Eye and Ear, Department of Ophthalmology, Harvard Medical School, Boston, MA, USA
REVIEW RETURNED	11-Jul-2020

GENERAL COMMENTS	The goal of the proposal is to (1) describe relationships between vision impairment and road crash risk and (2) to evaluate how interventions that restore vision reduce crash risk by reviewing existing literature. The protocol is well motivated and relevant. The methods of extracting, reporting, and synthesizing the articles are logical and based on many established guidelines. Below, I have listed a few of my comments: Page 4, Line 16: In the Strengths/Limitations section, does this mean that only open-access journals will be accessed? I may just be misunderstanding “open-access depository” Page 6, Line 29 and Page 10 lines 47-50: It is mentioned a few times about capturing data from those with vision impairment across the lifespan. I worry that many studies either use samples that are all older adults (as the authors note is most common with studies investigating vision impairment) or the sample is a range of ages and not the focus of the study. For example, in Wang, Moharrer, Baliutaviciute, Dougherty, Cybis, Bowers, & Luo 2020, naturalistic driving was recorded for individuals who use a bioptic telescope (also see Wood, McGwin, Elgin, Searcey, & Owsley
---

	2013). While this study seems relevant for this review, the participants' ages ranged from 22 to 90 and no analyses were done on their age. When referring to "across the lifespan", is the goal to include studies like those referenced above (which presumably would be excluded from studies looking just at older adults), and if so, how would their results be synthesized with studies that do focus on differences in age (e.g. Bowers, Sheldon, DeCarlo, & Peli, 2016)? I do not see any exclusion criteria concerning age in the methods, so I presume the goal is to simply not restrict by age. Page 6, Line 12: The most recent Cochrane review (2014) failed to find any RCTs that met the inclusion criteria. I believe the authors should be explicit in the differences between this review and the Cochrane and why they believe they will be more successful in finding studies within the inclusion criteria Page 7, Line 5: It is mentioned later (Page 8, line 10-12) that scores from "on-road tests or 'naturalistic' in-vehicle monitoring" will be used as measures of driving safety. Does this mean that all on-road studies will be used? Including close track, which certainly involve real driving albeit with unrealistic scenarios (given lack of traffic, actual hazards, etc)? Further, would this also include situations where the driving performance is assessed via a trained evaluator in the car? Each likely has different measures of driving performance that range in severity (actual collisions, near collisions, evaluator interventions, estimated collisions, etc). I think it is important to be specific about what kinds of studies will be included and how their differences will be mitigated Page 7, Line 8 and page 8, lines 24-26: Articles may be found by checking the reference list. Will the authors also look in the other direction, i.e., at the citation list? That is, if I published paper X 2020 that references paper Y 2018 and you have identified paper Y 2018 as relevant, could paper X 2020 be found because it cites paper Y 2018? I mostly make this suggestion to perhaps improve the likelihood of finding more studies. Page 7, Lines 14-25: While there is certainly debate about the validity of driving simulation, I currently do not see anything about simulated vision impairment, which is sometimes used in the literature (e.g., Wood & Troutbeck, 1994). Will simulated vision impairment also be screened and omitted from inclusion? Page 7, Lines 43-46: Do studies necessarily need to include an intervention to restore or improve vision to be included in this review? Its not clear whether the intervention is necessary, especially since it doesn't seem necessary for Aim 1 (i.e., describe the associations between vision impairments and risk of road crash involvement across the lifespan) and the need for the intervention likely reduces the number of included studies
--	---

VERSION 1 – AUTHOR RESPONSE

Reviewer 1

This protocol describes the method of a systematic review and meta-analysis that will investigate the relationship between vision impairment and MVC involvement, as well as the effectiveness of certain

interventions to reduce the rates of MVC involvement among vision-impaired drivers. The question is interesting and worthy of consideration. In addition, the study has a number of strengths – including its use of an on-road driving test. However, several important points remain to be clarified.

Title:

I find the “of associations” part of the title to be unclear. I would suggest rephrasing.

Thank you for the suggestion. The title has now been amended to:

“Associations between vision impairment and driving and the effectiveness of vision-related interventions: protocol for a systematic review and meta-analysis”

Abstract:

“Driving is becoming one of the main modes of transport” – is ‘becoming’ accurate here? Is it not already a very popular mode of transport? ‘However’ is also used incorrectly in this sentence, e.g. “with safe driving requiring a combination of...”

Thank you for the corrections. The sentence now reads:

“Driving is one of the main modes of transport with safe driving requiring a combination of visual, cognitive, and physical skills.”

Strengths and Limitations Section:

The first dot point is quite wordy. I would suggest refining this.

The point has now been edited to read:

“Results from this systematic review will present up-to-date evidence for the influence of vision impairment on road traffic injuries (RTIs) and the effectiveness of vision-related interventions.”

I would be cautious about considering the lack of a time constraint on your search as a strength. It is helpful in terms of providing a complete picture of the evidence; however, public health interventions and driving conditions have evolved a lot over the years and so findings from early studies might have limited relevance today.

Thank you for picking this up. The second point has been amended to reflect this:

“As there are no geographic restrictions in the criteria for included studies, this review will capture a large portion of English-language publications on this area with findings applicable to a global context.”

The second dot point is also quite vague. Increased data capture is obviously important – you need to focus on the implications of this (e.g. generalisability of findings). You could also incorporate something about the benefits of including all age brackets here.

Thank you for the suggestions. The second point has been amended as above to further explain the implications.

An additional point has been added to explain the benefits of including all age brackets:

“This review will not restrict the age of the target population allowing evidence on the impact of vision impairment and driving to be documented for all age groups.”

Will this review really exclude studies that are not published open access, or has this been described incorrectly? This method would result in a lot of missing data and be a significant limitation of the review. Please clarify.

Thank you for picking this up. This limitation was described incorrectly and has been amended:

“This review only looks at published studies in English, so research from non-English speaking countries may be missed, which could introduce bias.”

Introduction:

The introduction is informative and generally well written. However, I have noted a few minor issues below:

In the first paragraph, the authors describe the prevalence of RTI fatalities in LMIC. If I understand correctly, this has been done to provide insight into the extent of the issue and highlight the fact that RTI is a global problem. I appreciate the detail; however, it leaves the reader with the expectation that the review is going to focus on LMIC. I would suggest removing the third sentence about ‘growing disparities’ and perhaps replacing this with some additional detail about the state of the problem in high-income countries (even if it is less severe and improving).

Thank you for the suggestions, the sentence on LMIC disparities has been replaced with a sentence providing more context on HICs:

“Introduction”, paragraph 1, page 4:

“Individuals from lower socioeconomic backgrounds living in HICs are also more likely to be involved in a road crash resulting in injuries.”

I would suggest not starting the second sentence with the phrase “it has long been established”. I am not sure the second sentence of the second paragraph makes sense. How has the knowledge that MVCs are multifactorial evolved to the Safe System Approach? I would suggest rephrasing.

Thank you for the advice, this section of the second paragraph has been amended to explain the evolution of the Safe System Approach in more detail:

“Introduction”, paragraph 2, page 4:

“Using the Haddon Matrix, an early theory describing the multifactorial nature of RTIs, MVCs are understood to involve host (human), agent (vehicles and equipment), and environmental (physical and socioeconomic) factors(6). This theory has since been used to build the Safe System Approach endorsed in the United Nations Road Safety Collaboration’s Decade of Action for Road Safety (2011-2020).(7) In brief, the Safe System Approach aims to prevent MVCs which result in serious injuries or death by addressing four main pillars of focus: 1) safe roads, 2) safe speeds, 3) safe people and 4) safe vehicles.(8)”

References, Page 11:

8 Towards Zero Foundation. The Safe System; <http://www.towardszerofoundation.org/thesafesystem/>

(accessed 30 July 2020)

The first sentence of the third paragraph is missing some detail. The authors state that “standards [mostly for visual acuity] have been set” – can you clarify, in which regions are these enforced? (Or, if they are global, state this?). Also, based on your follow up sentences, I would assume that different regions use different methods to identify vision-impaired drivers. Could you perhaps note this (even just subtly in the way the first or second sentence is structured)? It might help to provide context and also introduce the fact that the effectiveness of the different interventions appears to vary and therefore warrants investigation.

Thank you for the suggestions, the beginning of the third paragraph has now been edited to provide more explanation on the jurisdictional control of vision and driving.

“Introduction”, paragraph 3, page 5:

“Due to the high visual demands needed to drive safely, many countries have set federal and or state-specific standards for vision, mostly for visual acuity. Most countries accept that a visual acuity of at least 6/12 (0.50, 20/40) in the better eye as the requirement for driving. This threshold dictates jurisdictional control used to identify individuals with vision impairment and restrict their access to driving privileges. However, a systematic review by Dobbs (2008) suggested that licencing policies aimed at identifying at-risk older drivers may not be effective in decreasing crash rates.(12) This may be because policies which govern licensure and vision screening vary significantly between and within countries. Further, evidence on their effectiveness is inconclusive.(13)”

References, Page 11:

Reference number 16 is now reference number 13:

13 Desapriya E, Harjee R, Brubacher J, et al. Vision screening of older drivers for preventing road traffic injuries and fatalities. *Cochrane Database Syst Rev* 2014(2):Cd006252. doi: 10.1002/14651858.CD006252.pub3

Could you perhaps give some examples of different vision screening procedures toward the end of the third paragraph?

Thank you for picking this up, examples of vision screening procedures have now been added:

“Introduction”, paragraph 3, page 5:

“A Cochrane review, updated twice, examined the benefits of different vision screening procedures, such as visual acuity, visual field (central or peripheral), contrast sensitivity, and useful field of view (UFOV) tests, in randomised-controlled trials (RCTs) aimed at preventing RTIs and fatalities in older drivers.(16, 17)”

I find it a bit unusual that the abstract emphasises the need for effective vision-related interventions due to the aging population, but that the aging population isn't considered in the introduction. Is this worthwhile noting?

Thank you for the suggestion. As this systematic review will aim to investigate the influence of vision impairment on driving in all ages, the ageing population was only mentioned in the abstract as context. However upon review, the end of the third paragraph in the introduction has now been amended to include a brief explanation on the importance of driving for older adults:

“Introduction”, paragraph 4, page 5:

“Since older drivers have higher crash involvement(20) and greater prevalence of eye diseases due to natural age-related declines in vision,(21) most research investigate older drivers and their risks of crashes and injuries. However, it is important to document the impact of vision impairment across all age groups. Further, information is also needed about specific eye diseases and types of vision impairment to inform interventions to screen for poor vision in drivers, and interventions to rehabilitate vision, thereby enhancing driver safety and continued ability to drive. This is especially important for older adults who rely on driving to remain independent and connected with their community. The loss of the ability to drive and the eventual retirement from driving has been linked to higher symptoms of depression and poorer health in older adults.(22)”

References, Page 12:

22 Ragland DR, Satiriano WA, MacLeod KE. Driving cessation and increased depressive symptoms. *J Gerontol A Biol Sci Med Sci* 2005;60(3):399-403. doi: 10.1093/gerona/60.3.399

Method:

The authors state that studies using driving simulators will be excluded. I assume similar studies using on-road assessment techniques will also be excluded. Perhaps state this outright.

Thank you for the comment. On-road assessments will be included but was not made clear in the original manuscript. A statement on their inclusion has been added to the beginning of the “Eligibility Criteria” section.

Paragraph 2, page 6:

“To obtain data on driving scores and performance, studies using on-road driving assessments, which include closed-circuit routes and those combining both closed and real-road driving tracks, and naturalistic driving with in-vehicle monitoring will be included.”

Studies will obviously have to have measured the primary outcome variable (MVC involvement) to be included in this review. I wonder whether this is worth noting in the ‘Eligibility Criteria’ section of your paper (and then referring the reader to the ‘Outcome Measures’ section of the manuscript for further detail)? I think you could use this information to strengthen your justification for excluding driving simulator studies. Currently, your comments on these studies appear to be tacked on the end of a paragraph about study design. I think it would be more appropriate to indicate that eligible studies had to have measured the primary outcome variable and that those driving simulator and on-road studies were excluded because they are not suitable for measuring MVC involvement, specifically? It would also be a more appropriate place to note that you did not intend to include studies investigating “self-regulatory behaviours” (again, specific outcomes).

Thank you for the suggestion, the second paragraph in the “Eligibility Criteria” section has been amended to better explain the studies to be included:

Paragraph 2, page 6:

“All studies must report on at least one of the outcome variables, described in the following section, which include MVC involvement and surrogate measures of driving safety such as driving errors and performance scores and driving cessation. Studies investigating either self-regulatory behaviours, such as night driving avoidance and decreasing travel mileage, or self-reported measures of driving safety, will be excluded. To obtain data on driving scores and performance, studies using on-road

driving tests, which include closed-circuit routes and those combining both closed and real-road driving tracks, and naturalistic driving with in-vehicle monitoring will be included. Even though closed-circuits may not reflect true on-road driving conditions, tests for common driving maneuvers such as road signage recognition, hazard recognition and avoidance, reversing, and gap perception are able to be recreated on these routes.(24) Driving errors and driving performance scores on the on-road driving tests can come from fitted in-vehicle monitoring technologies or trained observers. Studies which used driving simulators will be excluded as these are laboratory studies with only indirect measures of driving performance. MVC involvement cannot be measured and real-life driving experiences, such as limited exposure to driving at night, in bad weather, or during rush hour, may not be reflected in a simulation.(25) Additionally, the validity of driving simulator results are highly dependent upon the type of simulation program used and what kind of driving manoeuvre is being investigated.(26) As this review is interested in the MVC involvement and driving abilities of individuals who drive and their habitual vision, studies which simulate impairments in vision will also be excluded.”

References, page 13:

24 Wood JM. Age and visual impairment decrease driving performance as measured on a closed-road circuit. *Hum factors*. 2002;44(3):482-94. doi: 10.1518/0018720024497664

The authors plan to exclude studies of drivers with “specific medical conditions” and give examples of some conditions, most of which would significantly impact an individual’s ability to safely operate a motor vehicle. I wonder how they intend to handle studies of drivers with less overtly impairing medical conditions (e.g. diabetes)

Thank you for the comment, examples have been added. Studies will consider less overtly impairing medical conditions, such as diabetes, cardiovascular disease, and musculoskeletal conditions, as they occur in the general driving population. If these conditions are found to influence the driving outcomes of interest, the final systematic review will report on these results. This has been added to the “Eligibility Criteria”:

“Eligibility Criteria”, paragraph 3, page 7:

“Studies of drivers who have specific medical conditions (e.g. dementia, epilepsy, stroke, and history of medical events such as syncope)....”

Check grammar on Page 7 Line 33 (abstract or abstracts?).

Thank you for picking this up, the sentence now reads:

“Data Management and Selection”, page 8:

“Each title and abstract will be screened by two investigators...”

The authors state that “heterogeneity for all included studies will be assessed clinically, methodologically and statistically”. Can heterogeneity be assessed statistically for all included studies? If I understand correctly, this means that all included studies will need to have reported sufficient data to facilitate the completion of these analyses. If this is the case, it needs to be included as an eligibility criterion.

Thank you for the comment, yes the analyses will only be conducted if there is sufficient data in the included studies. We have planned to investigate heterogeneity across the studies therefore statistical heterogeneity will be assessed using subgroup meta-analysis and meta-regression. The second paragraph in the “Data Synthesis” section has now re-phrased to better explain the analysis plan for heterogeneity:

“Data Synthesis”, paragraph 2, page 9:

“Heterogeneity across all included studies with sufficient data will be assessed clinically, methodologically and statistically... Statistical heterogeneity across studies will be explored by formal statistical test of heterogeneity, subgroup analysis, and if feasible, by meta-regression.

I see now that the authors plan to meta-analyse several different outcomes. My initial interpretation was that MVC involvement was the only outcome that would undergo meta-analysis, since systematic reviews often draw their secondary outcomes only from studies that report the primary outcome – making them unsuitable for meta-analysis (i.e. since some data is not captured). It would be helpful if the authors could specify exactly what a study needs to have measured in order to be eligible for review.

Thank you for the suggestion, studies do not have report on MVC involvement to be included in the review. Studies reporting on only driving cessation or driving errors and performance scores will also be included. This inclusion criteria has been clarified in the amended “Eligibility Criteria” section described above:

Paragraph 2, page 6:

“All studies must report on at least one of the outcome variables, described in the following section, which include MVC involvement and surrogate measures of driving safety such as driving errors and performance scores and driving cessation.”

In regards to the meta-analysis, we plan to meta-analyse all outcomes we deem feasible to do so according to the available data. This has been clarified in the “Data Synthesis Strategy” section:

Paragraph 3, Page 9:

“The following outcomes will be assessed using meta-analysis where feasible according to data availability: crash involvement, driving cessation...”

Discussion:

Please review your grammar in the first sentence of the discussion; I'm not sure how this systematic review will care for people who would like to drive.

Thank you for picking this up, the sentence has been amended to read:

Paragraph 1, page 10:

“The findings of this systematic review may influence future road safety policies on driving and care for drivers with vision impairment.”

Please review the wording of the second sentence of the discussion. I think 'identifying' is the wrong term here. Perhaps "gaining a better understanding of visual factors that influence crash involvement" (or similar)?

Thank you for the suggestion, the sentence now reads:

Paragraph 1, page 10

"By understanding the visual factors contributing to MVCs, vision-related screening tests for licencing may be reconsidered and updated to increase relevance to driving safety."

Again, please review the wording of the last paragraph of the discussion. This doesn't seem to be particularly well-written/considered.

Thank you for the feedback. The last paragraph of the "Discussion" has now been amended for clarification:

Paragraph 3, page 11:

"Results from this review may also provide additional evidence on the impact of eye-disease specific interventions on quality of life factors, especially those related to driving and the ability to drive. Interventions to improve and optimise vision are needed for drivers, in recognition of the important of continued safe driving. This greater awareness will in turn also provide evidence for policies around road safety for individuals with vision impairments."

Reviewer 2

The goal of the proposal is to (1) describe relationships between vision impairment and road crash risk and (2) to evaluate how interventions that restore vision reduce crash risk by reviewing existing literature. The protocol is well motivated and relevant. The methods of extracting, reporting, and synthesizing the articles are logical and based on many established guidelines. Below, I have listed a few of my comments:

Page 4, Line 16: In the Strengths/Limitations section, does this mean that only open-access journals will be accessed? I may just be misunderstanding "open-access depository"

Thank you for pointing this out, the original manuscript had incorrectly described this limitation as studies not yet available in open-access depositories will be used. The point has now been amended and reads:

"This review only looks at published studies in English, so research from non-English speaking countries will be missed, which could introduce bias."

Page 6, Line 29 and Page 10 lines 47-50: It is mentioned a few times about capturing data from those with vision impairment across the lifespan. I worry that many studies either use samples that are all older adults (as the authors note is most common with studies investigating vision impairment) or the sample is a range of ages and not the focus of the study. For example, in Wang, Moharrer, Baliutaviciute, Dougherty, Cybis, Bowers, & Luo 2020, naturalistic driving was recorded for individuals who use a bioptic telescope (also see Wood, McGwin, Elgin, Searcey, & Owsley 2013). While this study seems relevant for this review, the participants' ages ranged from 22 to 90 and no analyses were done on their age. When referring to "across the lifespan", is the goal to include studies like those referenced above (which presumably would be excluded from studies looking just at older adults), and if so, how would their results be synthesized with studies that do focus on differences in

age (e.g. Bowers, Sheldon, DeCarlo, & Peli, 2016)? I do not see any exclusion criteria concerning age in the methods, so I presume the goal is to simply not restrict by age.

Thank you for comments. As this systematic review aims to capture studies looking at driving in individuals of all ages, we have not restricted age and will look at all age brackets which will include studies like the two you have pointed out. The “Eligibility Criteria” section has now been amended to better explain this:

Paragraph 3, page 7:

“The population of focus will be drivers of four-wheeled motorised vehicles such as cars, buses, and trucks. Unlike the two Cochrane reviews mentioned above which only focused on older drivers, this review will include drivers of all ages.”

As noted in the introduction, most research on vision impairment and driving focuses on older drivers. To synthesise the results, age will be assessed by means of specific subgroup analysis and/or meta-regression to possibly explain the heterogeneity across the studies. This has been clarified in the “Data Synthesis Strategy”:

Paragraph 3, page 9:

“As there is no age restriction on the focus population, results on age will be synthesised by assessing specific subgroup analysis and/or meta-regression which may partially explain heterogeneity across studies in the pooled effect size.”

Page 6, Line 12: The most recent Cochrane review (2014) failed to find any RCTs that met the inclusion criteria. I believe the authors should be explicit in the differences between this review and the Cochrane and why they believe they will be more successful in finding studies within the inclusion criteria

Thank you for the suggestion. The “Eligibility Criteria” has been amended to state the differences between this review and the previous Cochrane reviews:

Paragraph 1, page 6:

“Unlike the two previous Cochrane reviews which only included RCTs, this review will consider both interventional (RCTs and quasi-experimental) and observational (cohort, cross-sectional, and case-control) studies.”

Paragraph 3, page 7:

“Unlike the two Cochrane reviews mentioned above which only focused on older drivers, this review will include drivers of all ages.”

Page 7, Line 5: It is mentioned later (Page 8, line 10-12) that scores from “on-road tests or ‘naturalistic’ in-vehicle monitoring” will be used as measures of driving safety. Does this mean that all on-road studies will be used? Including close track, which certainly involve real driving albeit with unrealistic scenarios (given lack of traffic, actual hazards, etc)? Further, would this also include situations where the driving performance is assessed via a trained evaluator in the car? Each likely has different measures of driving performance that range in severity (actual collisions, near collisions, evaluator interventions, estimated collisions, etc). I think it is important to be specific about what kinds of studies will be included and how their differences will be mitigated

Thank you for the advice. On-road studies including closed-circuits and those that use a combination of a closed-track and real road sections will be included. The inclusion of closed-circuits is that these circuits have been created to reflect natural on-road driving conditions with sections that require participants to perform manoeuvres such as signage recognition, and hazard detection and avoidance. The “Eligibility Criteria” has been edited to better explain this point:

Paragraph 2, page 6-7:

“To obtain data on driving scores and performance, studies using on-road driving assessments, which include closed-circuit routes and those combining both closed and real-road driving tracks, and naturalistic driving with in-vehicle monitoring will be included. Even though closed-circuits may not reflect true on-road driving conditions, tests for common driving maneuvers such as road signage recognition, hazard recognition and avoidance, reversing, and gap perception are able to be recreated on these routes.(24)”

References, Page 13:

24 Wood JM. Age and visual impairment decrease driving performance as measured on a closed-road circuit. *Hum factors*. 2002;44(3):482-94. doi: 10.1518/0018720024497664

Thank you for pointing out the subjectiveness of the measure of driving performance. We anticipate that each study will have their own system for scoring and rating driving maneuvers. To overcome this, the investigators will decide upon a pass/fail threshold to be used for this review in order to compare the results from the included studies. The “Eligibility Criteria” and “Outcome Measures” sections has been amended to reflect this:

“Eligibility Criteria”, paragraph 2, page 7:

“Driving errors and driving performance scores on the on-road driving tests can come from fitted in-vehicle monitoring technologies or trained observers.”

“Outcome Measures”, paragraph 2, page 8:

“To account for differences in the criteria used by trained observers to evaluate the driving performance scores on on-road driving tests, a pass/fail threshold for driving performance scores specific to this review will be decided upon by all investigators in order to synthesise results.”

Page 7, Line 8 and page 8, lines 24-26: Articles may be found by checking the reference list. Will the authors also look in the other direction, i.e., at the citation list? That is, if I published paper X 2020 that references paper Y 2018 and you have identified paper Y 2018 as relevant, could paper X 2020 be found because it cites paper Y 2018? I mostly make this suggestion to perhaps improve the likelihood of finding more studies.

Thank you for the advice. This alternative way of capturing papers has now been added to the “Search Strategy” [page 8]:

“Additional potentially relevant studies will be sought by experts in the field by checking the reference lists and citations of included studies, and checking the reference list of narrative systematic reviews identified in the search.”

Page 7, Lines 14-25: While there is certainly debate about the validity of driving simulation, I currently

do not see anything about simulated vision impairment, which is sometimes used in the literature (e.g., Wood & Troutbeck, 1994). Will simulated vision impairment also be screened and omitted from inclusion?

Thank you for this suggestion. This review will not include papers simulating vision impairments as we are only interested in looking at the driving performance and outcomes of individuals who currently drive and their habitual vision. This reason has been made clearer in the “Eligibility Criteria”:

Paragraph 2, page 7:

“As this review is interested in the MVC involvement and driving abilities of individuals who drive and their habitual vision, studies which simulate impairments in vision will also be excluded.”

Page 7, Lines 43-46: Do studies necessarily need to include an intervention to restore or improve vision to be included in this review? Its not clear whether the intervention is necessary, especially since it doesn’t seem necessary for Aim 1 (i.e., describe the associations between vision impairments and risk of road crash involvement across the lifespan) and the need for the intervention likely reduces the number of included studies

Thank you for pointing this out. Studies do not have to report on an intervention in order to be included in the review as mentioned in the beginning of the “Eligibility Criteria” which explained our intention to include interventional and observational studies. This point has been re-iterated for clarification at the end of the “Eligibility Criteria”:

Paragraph 4, page 7:

“Even though it is not necessary for all included studies to report on vision-related interventions, studies which do report on interventions can include procedures such as vision screening, refractive correction, cataract surgery or other measures to restore and improve vision of drivers in order to maintain driving participation, promote safe driving and reduce risk of crash involvement.”

VERSION 2 – REVIEW

REVIEWER	Danielle McCartney University of Sydney
REVIEW RETURNED	28-Aug-2020

GENERAL COMMENTS	The second dot point is unclear. The authors state that “as there are no geographic restrictions..., this review will capture a large portion of English-language publications on this area” – perhaps it would be more accurate to say ‘in this research area?’ Thank you for providing more information on the outcomes you plan to assess. I wonder now (knowing the outcomes you plan to investigate), why driving simulator studies will be excluded? I understand that they only provide an indirect measure of driving performance, but I would think they could provide useful insight into things like road signage recognition, hazard recognition and avoidance, etc.? I would suggest providing a stronger justification for their exclusion (or considering including these studies). Please again review the first sentence of the discussion; I don’t believe ‘care’ is the correct term here.
--

REVIEWER	Garrett Swan Schepens Eye Research Institute
REVIEW RETURNED	08-Sep-2020

GENERAL COMMENTS	The authors have done a good job revising the manuscript according to my comments and the comments from the other reviewer. The manuscript reads better now and the methodology is clearer than before. I do have a minor comment. Page 6, Line 36 in the cleaned copy of the paper – It says, “All studies must reported on at least one of the outcome variables” with driving cessation being one of these variables. However, in the next sentence, it says that studies investigating “self-regulatory behaviors, such as night driving avoidance and decreasing travel mileage” will be excluded. What is the specific reason for omitting self-regulatory behavior studies? I presume some information about the sample’s driving characteristics (e.g., how far/frequently they drive), including if they regulate their own driving, is important in understanding the differences between the sample. Will these factors be included in the participant characteristics? On Line 18 page 9, only demographic and vision details are described as participant characteristics.
---

VERSION 2 – AUTHOR RESPONSE

Reviewer 1

The second dot point is unclear. The authors state that “as there are no geographic restrictions..., this review will capture a large portion of English-language publications on this area” – perhaps it would be more accurate to say ‘in this research area?’

Thank you for the correction. The dot point as now been amended.

“Strengths and Limitations”, point 2, page 3:

“As there are no geographic restrictions in the criteria for included studies, this review will capture a large portion of English-language publications in this research area with findings applicable to a global context.”

Thank you for providing more information on the outcomes you plan to assess. I wonder now (knowing the outcomes you plan to investigate), why driving simulator studies will be excluded? I understand that they only provide an indirect measure of driving performance, but I would think they could provide useful insight into things like road signage recognition, hazard recognition and avoidance, etc.? I would suggest providing a stronger justification for their exclusion (or considering including these studies).

Thank you for the suggestion. We agree that driving simulator outcome measures like sign recognition and other measures are of interest in this field of research, however this is outside the scope of this current review. This review will be restricted to investigating direct measures of driving and driving cessation. This has been further clarified in the “Eligibility Criteria”:

Paragraph 2, page 6:

“To restrict the scope of the study to direct measures of driving, studies which used driving simulators will be excluded as these are laboratory studies with only indirect measures of driving performance.”

Please again review the first sentence of the discussion; I don't believe 'care' is the correct term here. Thank you for pointing this out. The first sentence has been edited to better reflect the review and now reads:

"Discussion", paragraph 1, page 10:

"The findings of this systematic review may influence future road safety and licencing policies on driving for drivers with vision impairment."

Reviewer 2

Page 6, Line 36 in the cleaned copy of the paper – It says, "All studies must reported on at least one of the outcome variables" with driving cessation being one of these variables. However, in the next sentence, it says that studies investigating "self-regulatory behaviors, such as night driving avoidance and decreasing travel mileage" will be excluded. What is the specific reason for omitting self-regulatory behavior studies? I presume some information about the sample's driving characteristics (e.g., how far/frequently they drive), including if they regulate their own driving, is important in understanding the differences between the sample. Will these factors be included in the participant characteristics? On Line 18 page 9, only demographic and vision details are described as participant characteristics.

Thank you for the suggestion. While self-regulatory behaviours are of interest, they are outside the scope of this review. We are including driving cessation as an outcome measure because, as mentioned in the Introduction, it has been linked to negative health impacts and depression in older adults.

Driving characteristics including mileage and licence status will be included in the participant characteristics where available. This change has been made in "Data Synthesis Strategy":

Paragraph 2, Page 9:

"Clinical heterogeneity will be accessed by comparing the differences between the participant characteristics (e.g. age, sex, eye disease, driving mileage, licence status or other available measures of driving exposure), interventions and outcomes measured."